# ACHIEVING EXACT FEDERATED UNLEARNING WITH IMPROVED POST-UNLEARNING PERFORMANCE

## ABSTRACT

Federated learning is a machine learning paradigm that allows multiple clients to train aggregated model via sharing model updates to a central server without sharing their data. Even though the data is not shared, it can indirectly influence the aggregated model via the shared model updates. In many real-life scenarios, we need to completely remove a client's influence (unlearning) from the aggregated model, such as competitive clients who want to remove their influence from the aggregated model after leaving the coalition to ensure other clients do not benefit from their contributions. The influence removal is also needed when the adversarial client negatively affects the aggregated model. Though the aggregated model can be retrained from scratch to ensure exact unlearning (completely removing the client's influence from the aggregated model), it performs poorly just after the unlearning, which is undesirable during deployment. To overcome this challenge, this paper proposes federated unlearning algorithms that ensure exact unlearning while achieving better performance post-unlearning. Our experimental results on different real datasets validate the performance of the proposed algorithms.

## 1 INTRODUCTION

An individual user may have insufficient data to train a state-of-the-art machine learning model. Yet, we can significantly improve the model performance by leveraging the combined data from multiple users. *Federated learning* (FL) (Zhang et al., 2021) is one of the most prevalent paradigms to perform such collaboration today, especially in sectors with strong privacy demands such as finance and health care (Li et al., 2020; Xu et al., 2021). In the FL setting, collaborative clients train local models on their own data, and a central server model is obtained by aggregating these local model updates for multiple communication rounds. FL is well-suited for many commercial applications as it eliminates the need to share users' private data during training. For example, multiple companies from the same industrial sector (e.g., banking, insurance, or healthcare) often possess diverse user data. To leverage all available data, these companies can collaborate and train a more accurate model using suitable FL algorithms (Aledhari et al., 2020).

Although FL algorithms do not directly access users' data, the aggregated model is still influenced by the local models trained on each client's data. When a client leaves the collaboration, it is necessary to update the aggregated model to remove the influence of its data–a process known as *federated unlearning* (FU),[1] e.g., a company leaving the collaboration of many companies may demand the removal of their contributions to ensure their competitors do not benefit from them. FU techniques are also desirable to remove the influence of adversarial clients, i.e., the adversary behaves like a client and degrades the model performance by contributing contaminated updates (Fang et al., 2020). Additionally, the development of FU techniques facilitates the exercise of the *right to be forgotten* formalized in many regional or government data regulations such as GDPR (2016) and CCPA (2018).

We can trivially achieve FU by retraining the collaboration from scratch without the target client's data (Liu et al., 2023). Despite its simplicity, the new server model suffers from low performance as it is restarted with random initialization. As a result, it slows down the deployment of the unlearned model, as training large models on the collaboration of many users can be time-consuming. Due to

---

[1]This differs from the typical FL setting, where clients may be intermittently active or inactive during the training process.

these shortcomings, it is natural to consider the following question: *How can we guarantee the exact federated unlearning while ensuring better post-unlearning performance?*

This paper proposes two novel methods for achieving exact FU with improved post-unlearning performance. The first method, Bi-Models Training (BMT) (Section 3.1), preserves isolated copies of local models and reuses clients' existing knowledge residing in these models during unlearning for better aggregation. Despite being unlearning-friendly, these local models fail to capture the joint influence of multiple clients on the global model. Training the power set of clients can capture the influence of all possible influences of the clients but is computationally expensive and may lead to *double influence*, where a client affects multiple sub-FL models. As a result, we propose the second method, Multi-Models Training (MMT) (Section 3.2), that trains each sub-FL model on disjoint subsets of clients to avoid double influence and aggregates the best sub-FL models upon unlearning to achieve improved initialization of the aggregated model. We empirically justify the effectiveness of BMT and MMT through multiple experiments on real-world vision and language datasets (Section 4).

## 1.1 RELATED WORKS

In this section, we now review the relevant work, especially in federated learning, machine unlearning, and federated unlearning, to our problem setting.

**Federated Learning (FL).** FL emerges from the industrial needs to train centralized models on large, decentralized data residing on users' device (McMahan et al., 2017) and is particularly favored in sectors requiring strong privacy guarantees, such as finance and health care (Li et al., 2020; Xu et al., 2021). Based on the characteristics of the decentralized data, Yang et al. (2019) divided FL into three categories: horizontal FL, vertical FL, and federated transfer learning. To optimize the federated models, McMahan et al. (2017) proposed the FedAvg algorithm that averages local updates from contributing clients and works well on independent and identically distributed (i.i.d.) data. However, as real-world data is often heterogeneous (e.g., users with different demographics), subsequent works have proposed new methods that target model architecture or algorithm design to alleviate model drift that can degrade model performance (Zeng et al., 2023; Mu et al., 2023; Idrissi et al., 2021; Li et al., 2021; Karimireddy et al., 2020). We refer the readers to Zhang et al. (2021) for a detailed survey of various works covering different settings of federated learning.

**Machine Unlearning (MU).** MU aims to remove the influence of a selected subset of data from the trained ML model. Based on the guarantee of removal, MU methods are broadly categorized into exact unlearning and approximate unlearning (Nguyen et al., 2022; Wang et al., 2024). In exact unlearning, we aim for an identical model to one that would have been obtained by retraining without that data to be erased. Retraining is a method that trivially achieves exact unlearning but is computationally expensive with large models and datasets. Existing works can exactly unlearn for support vector machines (Cauwenberghs & Poggio, 2000), k-means (Ginart et al., 2019), random forests (Brophy & Lowd, 2021). Bourtoule et al. (2021) partitions the entire training data set into a few disjoint subsets and trains one base model with each of these subsets. Since each base model is only trained with a subset of the original training data, the performance may be sub-optimal. Approximate unlearning aims for a model whose distribution closely resembles that of the retrained model. Guo et al. (2020) proposed a certified removal method to approximately unlearn linear model by Newton-like update. Nguyen et al. (2020) minimizes the KL divergence between the approximate posterior of the unlearned model and the retrained model under the variational inference framework.

**Federated Unlearning (FU).** Many recent works adapt machine unlearning to the federated learning settings (Liu et al., 2020; Wang et al., 2021; Gong et al., 2021). Liu et al. proposed FedEraser, which involves using historical updates from the server and local calibration training on the unlearned client. The federated unlearning protocol proposed in this work can be used to unlearn an arbitrary subset of clients without any constraint on the type of data each client possesses. At the same time, it requires no participation of the unlearned client. Wang et al. proposed a channel pruning-based method to selectively forget a specific class from the trained ML model. Such an approach has limited scope as it is impractical to assume that each participant in the FL setting possesses precisely one class of data. Gong et al. concerned with the setting where no centralized party/server is present, which does not apply to the centralized FL setting. In terms of exact federated unlearning, Xiong et al. (2023) and Tao et al. (2024) use quantization and sampling strategies, respectively, to get a checkpoint during the FL training where the unlearned client's data have not made a quantifiable impact and use it

as initialization for model retraining and since speed up the retraining process. On the other hand, Qiu et al. (2023) proposed to cluster the clients and train a few intermediate FL models and then subsequently obtain the global FL model through one-shot aggregation. At the unlearning stage, only the intermediate FL model where the unlearning client is present is retrained (and hence reducing the retraining cost). Our proposed method touches on both ideas and uses aggregation of a few sub-FL models to obtain a good initialization for much more efficient retraining. The way we obtained our sub-FL models trades-off between computation budget and post-unlearning performance, played an essential role in ensuring its effectiveness.

## 2    PROBLEM SETTING

**Federated Learning.**    This paper considers the centralized federated learning (FL) setting with a trusted central server and multiple clients. In this setting, a central server shared an aggregated model with the clients and then each client trains this model on his dataset and send model updates (weights or gradients) to the central server, which then aggregates these updates to get a better aggregated model. In our setting, we assume that the number of clients participating in FL process varies over time. Let $\mathcal{C}_t$ denote the set of participating clients at the beginning of the FL communication round $t$. An FL communication round (communication round for brevity) represents one cycle of model sharing by the central server with clients and then receiving the updated aggregated model.

Each client $c \in \mathcal{C}_t$ has training dataset $\mathcal{D}_{c,t}$ with $n_{c,t}$ labeled samples, where each sample is drawn from the distribution $\nu_c$ over $\mathcal{X} \times \mathcal{Y}$. Here, $\mathcal{X}$ represents the input space, and $\mathcal{Y}$ represents the label space. The learning model is denoted by $h_\theta : \mathcal{X} \to \mathcal{Y}$ for model parameters $\theta \in \mathbb{R}^d$, where $d$ is the number of model parameters. The loss incurred by the learning model $h_\theta$ on a sample $(x, y) \in \mathcal{X} \times \mathcal{Y}$ is denoted by $l(h_\theta(x), y)$, which can be the root mean squared error (for regression problems) or cross-entropy loss (for classification problems).

After the communication round $t$, the loss incurred by the client $c$ for model parameters $\theta$ is the average loss of the model $h_\theta$ on the samples in $\mathcal{D}_{c,t}$ and defined by $f_{c,t}(\theta) := \frac{1}{n_{c,t}} \sum_{s=1}^{n_{c,t}} l(h_\theta(x_{c,s}), y_{c,s})$, where $(x_{c,s}, y_{c,s})$ is the $s$-th sample in $\mathcal{D}_{c,t}$. The central server aims to find a learning model with the minimum average loss for each client. The server achieves this by finding a model $\theta$ that minimizes the average clients' loss weighted by their respective number of samples, which is given by solving the following optimization problem in the communication round $t$:

$$\operatorname*{argmin}_{\theta} \frac{1}{n_t} \sum_{c \in C_t} n_{c,t} f_{c,t}(\theta) = \frac{1}{n_t} \sum_{c \in C_t} \sum_{s=1}^{n_{c,t}} l(h_\theta(x_{c,s}), y_{c,s}), \tag{1}$$

where $n_t = \sum_{c=1}^{C_t} n_{c,t}$. Since the clients cannot share their local data $D_{c,t}$ with the server (due to communication or privacy constraints), the optimization problem given in Eq. (1) must be solved in a federated manner by using the suitable FL algorithm (e.g., FedAvg (McMahan et al., 2017)).

**Exact Federated Unlearning.**    Let client $c$ influence be completely removed from the aggregated model. Exact federated unlearning is the process of completely removing the influence of client $c$ training data from the aggregated model, resulting in a model that is equivalent to the models trained without the training data of client $c$. However, the aggregated model resulting from retraining without the data of client $c$ may have a poor performance in the initial round, which may not be expected when these models are deployed in practice. Therefore, our goal is to design methods that ensure exact federated unlearning while leading to an aggregated model with as high accuracy as possible.

## 3    EXACT FEDERATED UNLEARNING METHODS

Due to multiple communication rounds of the FL training, it becomes impossible to completely remove a client's data influence from the trained aggregated model.  Therefore, the most straightforward way to achieve the exact federated unlearning is to restart the federated learning process from scratch with the remaining clients. This method of retraining the aggregated model from scratch is called *retraining from scratch* (RfS) (Bourtoule et al., 2021; Liu et al., 2023). Although RfS is a simple method, the new model may have very low accuracy in the initial rounds after unlearning compared to the aggregated model before unlearning due to restarting the FL process with

random initialization of the aggregated model. Such performance reduction of the aggregated model may not be desirable during deployment in practice involving critical applications such as healthcare (Prayitno et al., 2021; Dhade & Shirke, 2024) and finance (Long et al., 2020). This shortcoming of `RfS` raises a natural question: How can we guarantee the exact federated unlearning while ensuring better post-unlearning performance? To answer this question, we propose two novel methods for achieving exact federated unlearning that completely remove the client's influence while giving better post-unlearning performance than `RfS`.

### 3.1 BI-MODELS TRAINING

To get a better performing aggregated model post-unlearning, we must design a new FL training process that allows exact federated unlearning while having a better initialization than random initialization. One way to achieve better initialization is to design methods that can exploit the remaining clients' existing knowledge. To do this, we propose a method named Bi-Models Training (`BMT`) that can be incorporated into any existing federated learning framework. The main idea of `BMT` is to have an additional model for each client that is only trained on its data, making these models unaffected by other clients' training data. We refer to this model as *local model*. We use the term *global model* for referring to the aggregated model, which is trained using all client's data and used for deployment. Next, we discuss how `BMT` can be incorporated into the different stages of any existing federated learning framework (as depicted in Fig. 1), namely: Initialization, FL Training, Unlearning, and New Client joining the FL process, whose details are given as follows.

**Initialization.** The central server starts the standard FL training process by randomly initializing the global aggregated model. This randomly initialized global model is then shared with all clients. Each client updates the global model using its local training data and then shares the model update (updated model or gradients) with the central server. As compared to the standard initialization in any FL training process, each client makes a copy of the locally updated global model[2] (i.e., local model). Since the initial global model is randomly initialized, these local models are, by design, isolated from the influence of other clients' training data.

**FL Training.** After receiving the first model updates, the central server aggregates them to get the aggregated global model as per the underlying FL algorithm (McMahan et al., 2017; Shlezinger et al., 2020; Zhang et al., 2021). In each subsequent communication round, each client receives the updated global model from the central server and then trains it using its training data. After updating the global model, each client shares the model update with the central server. Besides the standard FL training process, each client also updates their local model using their training data.

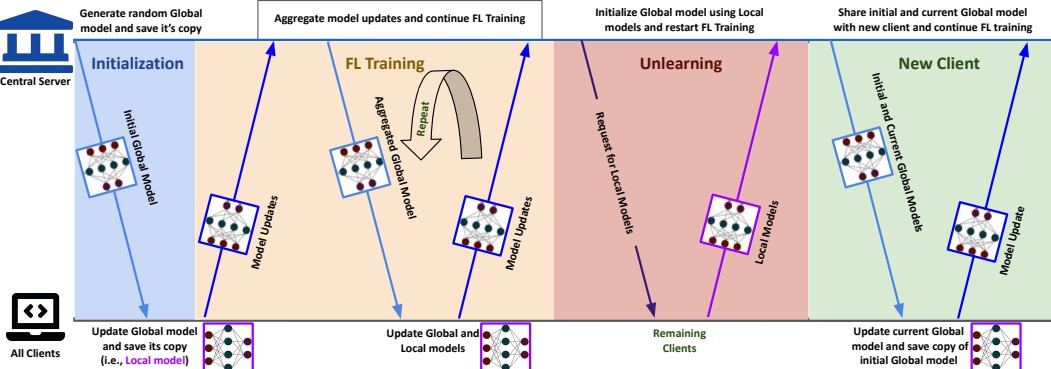

Figure 1: Bi-Models Training (`BMT`) in the different stages of any federated learning framework.

**Unlearning.** Let $c$ be the client whose influence needs to be completely removed from the global model after a communication round $t$ and $\mathcal{C}_{t,r}$ be the set of remaining clients, i.e., $\mathcal{C}_{t,r} = \mathcal{C}_t \setminus \{c\}$. The central server first discards the current global model and requests each client to share their current copy of local models. Once the central server receives the local models from all remaining clients, the central server aggregates them to get the new initialization for the global model as per the underlying

---

[2]The locally updated global model in the first communication rounds is the same as the model that is a copy of the initial global model and trained on client's training data.

FL algorithm, e.g., for FedAvg, the central server performs weighted aggregation on the remaining client's local models, where each client's weight is proportional to their respective training data. Our extensive experimental results (in Section 4) show that the resulting initialized global model performs better than random model initialization as done in RfS. Lastly, the central server restarts the FL training process with the newly initialized global model, which is completely free from the influence of the unlearned client's data.

**New Client.** When a new client wants to join the ongoing FL collaboration, the central server waits until the end of the ongoing communication round. Once it is over, the central server starts the FL training process with the new client by sharing the current global model with the new client, who then updates the current global model using its training data and shares the model update with the central server. Apart from this, the central client also shares the randomly initialized global model with the new client, who updates it, which then acts as the local model of the new client for subsequent rounds. Other clients do not influence this local model, as the initial global model is randomly initialized.

In summary, BMT has two models for each client: global and local. All clients train their local model on their data in isolation, whereas the global model is trained using the underlying FL training protocol. To completely remove a client influence from the global model, the central server first discards the global model and then uses the local models of the remaining clients to re-initialize the global model, which is further updated via FL training. This process ensures that BMT, by design, guarantees the exact federated unlearning. Further, using the remaining clients' local models leads to an initialization of the global model that is already influenced by the remaining clients to some extent, leading to a better performance than RfS, as corroborated by our experiments in Section 4. The price for this improved post-unlearning performance is the cost of pre-training the local models in advance. Such a trade-off is worthwhile for applications that require exact unlearning and an unlearned model with good performance as quickly as possible for deployment.

### 3.2 MULTI-MODELS TRAINING

The key insight of the previous section is that BMT achieves a better initial global model because it is influenced by the clients' local models. However, the local model only contains influence from an individual client and has no joint influence of multiple clients. Since all clients influence the global model, we should capture the joint influence of different clients and then use it to get a better initialization of the global model. To capture the joint influence, we can train FL models using only a subset of clients. We refer to these FL models as *sub-FL models*. Formally, a sub-FL model is an FL model that is trained via FL protocol using a subset of clients, where the size of the subset varies from 2 to $N - 1$. One can train all possible sub-FL models (power set of clients excluding global model) to capture the influence of all possible interactions of different subsets of clients. However, this approach is not computationally feasible as these sub-FL models increase exponentially with the number of clients (i.e., $2^n - n - 2$ for $n$ clients). Another problem of training arbitrary sub-FL models leads to a situation of *double influence*, which is defined as follows:

**Definition 1.** *Let $S_i$ be the set of clients whose data are used in training the $i$-th sub-FL model. The sub-models $i$ and $j$ leads to double influence if $S_i \cap S_j \neq \emptyset$, $S_i \setminus S_j \neq \emptyset$, and $S_j \setminus S_i \neq \emptyset$.*

When one client data is used to train two sub-FL models, it can lead to double influence if both are also trained using data from different clients, e.g., one is trained on clients $\{1, 2\}$ and another on clients $\{1, 3\}$; the client 1 data is used in both sub-FL models and hence having the double influence.

To avoid the double influence, each sub-FL model should be trained on disjoint subsets of clients, or the set of clients used for training sub-FL models is a proper superset of the set of clients used for another sub-FL model. One possible way to achieve this is to organize sub-FL models in a hierarchical tree structure. In this tree, the root node represents the global model while the leaf nodes correspond to the local models, and intermediate nodes represent sub-FL models, with each child node having disjoint sets of clients compared to its siblings. As we move from the root node to the leaf nodes, each sub-FL model branches into further subsets, maintaining either disjoint relationships or superset relations, thus ensuring a clear and systematic flow of influence throughout the hierarchy. We refer to this hierarchical tree structure as an *influence tree*. After unlearning a client, we should aggregate the sub-FL models with higher influence (those influenced by a larger number of clients) and local models to get the initialization for the global model. If the number of models to aggregate is less, it implies that the initialization of the global model contains the most joint influence of clients.

This relationship inspires our proposed metric *influence degradation score*, which measures how good is an influence tree. Next, we formally define the influence degradation score.

**Definition 2** (Influence Degradation Score (IDS)). *Let $\mathcal{T}$ be any influence tree structure. The influence degradation score for $\mathcal{T}$, denoted by $s(\mathcal{T})$, is defined as the average number of sub-FL and local models that are aggregated to get the initial global modal after unlearning any client.*

Though the tree structure, by design, eliminates double influence, we do not know which tree structure gives the lowest IDS for given clients' different likelihood of requesting unlearning (as the probability of requesting unlearning may vary across the clients). As our goal is to construct an influence tree with minimum IDS, our following result shows that the binary influence tree constructed using Huffman coding has the lowest IDS among all $n$-ary influence tree structures, where $n > 2$.

**Theorem 1.** *Given an $n$-ary influence tree $\mathcal{T}$, there exists a binary influence tree $\mathcal{T}_2$ that has smaller IDS, i.e., $s(\mathcal{T}_2) < s(\mathcal{T})$. Let $p_c$ be the unlearning probability of the client $c$. Then, Huffman coding with $n$ symbols representing clients and weights $\{p_c\}_{c=1}^n$ gives the optimal binary influence tree such that $s(\mathcal{T}_{Huffman}) \leq s(\mathcal{T}_2)$ for any influence tree $\mathcal{T}_2$ for the same group of clients.*

With Theorem 1, we can use Huffman coding (Huffman, 1952) to construct an influence tree that has the lowest IDS among all types of influence trees. In some real-life applications, the client's unlearning probability can be unknown. In such cases, we can assume that each client is equally likely to be unlearned, hence having the same unlearning probability. We show the influence tree for 8 clients having the same unlearning probability in Fig. 3a. A client (on the leaf node) influences the sub-FL model if there is a path from a sub-FL model to the leaf node representing that client. We next propose a method named Multi-Models Training (MMT) that uses the sub-FL models to get better initialization for the global model. MMT can be easily incorporated into the different stages of any existing federated learning framework (as depicted in Fig. 1), whose details are given as follows.

**Initialization.** Similar to BMT, the central server starts the standard FL training process by randomly initializing the global aggregated model. This randomly initialized global model is then shared with all clients. Each client updates the global model using its local training data and then shares the model update (updated model or gradients) with the central server. Each client makes a copy of the locally updated global model. Compared to BMT, MMT also initializes the sub-FL models using the model updates of clients corresponding to the sub-FL models.

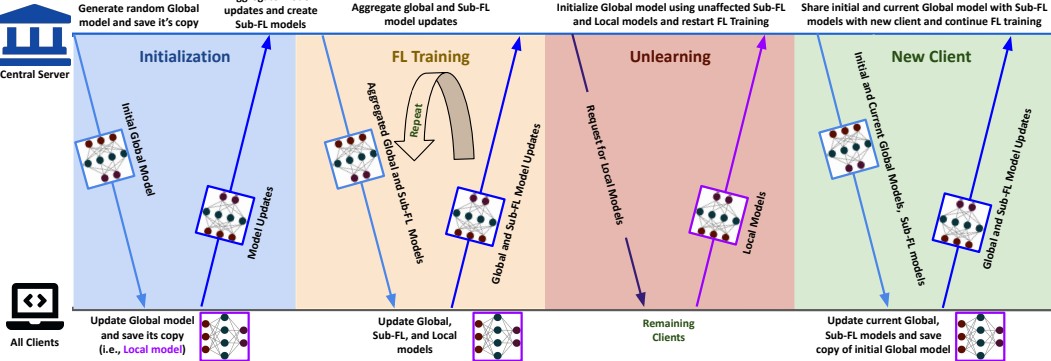

Figure 2: Multi-Models Training (MMT) in the different stages of any federated learning framework.

**FL Training.** After receiving the first model updates, the central server aggregates them to get the aggregated global and sub-FL models. In each subsequent communication round, each client receives the updated global and its sub-FL models from the central server and then trains them using its training data. After updating these models, each client shares the global and its sub-FL model updates with the central server. Apart from this, each client also updates their local model.

**Unlearning.** For unlearning a client, the central server first discards the current global model and related sub-FL models (as shown in Fig. 3b after unlearning client 2) and then requests all clients not in any of the remaining sub-FL models to share their current copy of local models. Once the central server receives all requested local models, it aggregates them with sub-FL models (choosing only the most influential unaffected sub-FL model over its descendants) to get the new initialization for the global model as per the underlying FL algorithm. After removing the sub-FL models related to the

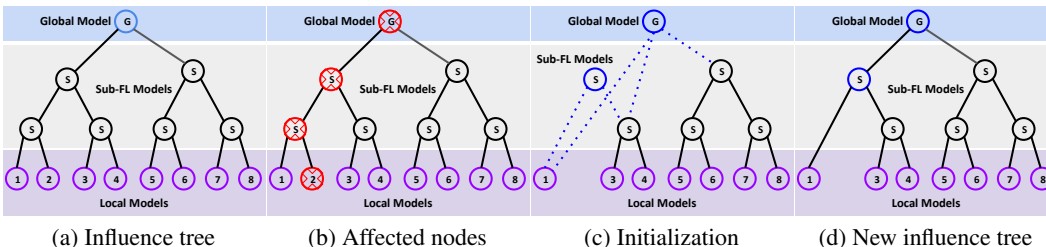

|                |                |                   |                     |
|:--------------:|:--------------:|:-----------------:|:-------------------:|
| (a) Influence tree | (b) Affected nodes | (c) Initialization | (d) New influence tree |

Figure 3: **Fig. 3a:** Influence Tree for 8 clients having the same unlearning probability. **Fig. 3b:** Showing the global and all sub-FL models affected after unlearning client 2 by the node's red cross and red outline. **Fig. 3c:** Initialization of global and new sub-FL models, where a dotted blue line shows the the models used to initialized them. **Fig. 3d:** Final influence tree after unlearning the client.

unlearned client, the remaining influence tree may no longer have the lowest IDS for the remaining clients. It leads to two options: create a new influence tree while using earlier sub-FL models as much as possible (as shown in Fig. 3c) or keep using the existing influence tree, which may not be the best but retains the sub-FL models trained over time. Lastly, the central server restarts the FL training process with the newly initialized global and sub-FL (if any) models (as shown in Fig. 3d), which are completely free from the influence of the unlearned client's data.

**New client.** Adding a new client to the ongoing FL collaboration can worsen the existing influence tree compared to the influence tree created using a new client. Like BMT, when a new client wants to join, the central server waits until the end of the ongoing communication round. Once it is over, the central server can create a new influence tree while using earlier sub-FL models as much as possible or keep using the existing influence tree to retain the existing sub-FL models, which are trained over time by adding new sub-FL models. After that, the central server starts the FL training process with the new client by sharing the current global and corresponding sub-FL models with the new client, who then updates the current models using its training data and shares the model updates with the central server. The central client also shares the randomly initialized global model with the new client, who updates it, which then acts as the local model of the new client for subsequent rounds.

Overall, the initialization of the global model in MMT has the joint influence of multiple clients, which makes it better than BMT and hence leads to better post-unlearning performance, as corroborated by our experiments in Section 4. However, note that there is an additional computational cost for this improved performance over BMT as we need to train multiple sub-FL models in parallel.

## 4 EXPERIMENTS

In this section, we empirically verify the effectiveness of the proposed methods in two important settings: (1) *sequential unlearning* setting, where multiple clients sequentially leave the federation, and (2) *continual learning and unlearning* setting, where clients can join and/or leave the federation at will. Subsequently, we analyze the impact of data heterogeneity and the branching factor in the MMT structure on the model performance. Then, we consider special scenarios when the clients follow a fixed unlearning order (e.g., according to their subscription plans) and clients with non-uniform unlearning probabilities (e.g., clients from different demographics).

### 4.1 EXPERIMENTAL SETTING

**Datasets.** We conduct our experiments on four popular vision datasets: MNIST (LeCun et al., 1998), FashionMNIST (Xiao et al., 2017), CIFAR-10 (Krizhevsky et al., 2009) and CIFAR-100 (Krizhevsky et al., 2009). We also consider language tasks with large language models in Section 4.5. To simulate clients with realistic non-IID data, we let the client $i$ receives the most data from the $i$-th class and the same amount of data from the remaining classes. We use $\rho$ to denote the ratio between the majority class and minority class for all clients. Each client contains 200 training/test samples and $\rho = 0.02$ for MNIST and FashionMNIST; 1000 training samples, 300 test samples, $\rho = 0.2$ for CIFAR-10; 400 training samples, 100 test samples, $\rho = 0.1$ for CIFAR-100.

**Models.** For MNIST and FashionMNIST, we use simple MLP networks with 30 and 80 hidden units, respectively. For CIFAR-10, we use a CNN network with $5 \times 5$ convolutional layers followed by $2 \times 2$ max pool layer for feature extraction and two fully connected layers with 32 hidden units and ReLU for classification. For CIFAR-100, we use a VGG-16 model (Simonyan, 2014).

**Training.** We use FedAvg (McMahan et al., 2017) to train FL models for 100 rounds with 1 local epoch on MNIST and FashionMNIST, 300 rounds with 1 local epoch on CIFAR-10 and 100 rounds with 10 local epoch on CIFAR-100. We use the SGD optimizer with a learning rate 0.01, weight decay 0.1, batch size 20, and gradient clipping 10 for MNIST and FashionMNIST. We use the AdamW optimizer with batch size 64 and the same hyperparameters for CIFAR-10. We use the SGD optimizer with a learning rate 0.005, momentum 0.9, weight decay $10^{-5}$, and batch size 64 for CIFAR-100. Our experiments are conducted on NVIDIA L40 46GB and NVIDIA H100 80GB GPUs.

**Metrics.** We report test accuracy measured on a fixed test set that combines local test sets of all possible clients, including those that join/leave the federation in later stages. The combined test set allows us to observe the visible trend of the performance after one client is removed.

**Comparison Methods.** We compare `BMT` and `MMT` against the following baselines: Standalone, where the centralized model trains on aggregated data from all remaining clients; Retraining from Scratch (`RfS`), where the federated model is retrained excluding data from the leaving client; FedCIO (Qiu et al., 2023); Exact-Fun (Xiong et al., 2023); and FATS (Tao et al., 2024).

## 4.2 SEQUENTIAL UNLEARNING

This experiment simulates a practical scenario when clients gradually leave the federation. After each client leaves, we continue training the federated model on the remaining clients. Particularly, we simulate the leaving of 3 clients $\{1, 3, 5\}$. It is noteworthy that this unlearning order is to `MMT`'s disadvantage as none of the sub-FL models can completely replace the server model. Hence, the server parameters must be aggregated from the parameters of other sub-FL models.

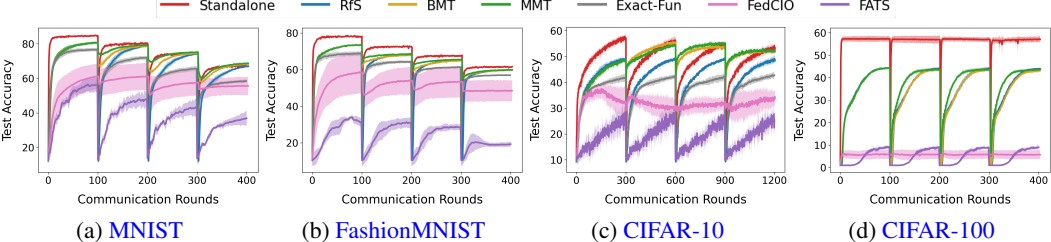

(a) MNIST  (b) FashionMNIST  (c) CIFAR-10  (d) CIFAR-100

Figure 4: Test accuracy in the sequential unlearning setting for different datasets.

Fig. 4 shows performance in the sequential unlearning setting on different datasets. As can be seen, `BMT` and `MMT` consistently outperform other methods by a large margin with better initialization and faster convergence after unlearning[3]. These results highlight the effectiveness of `BMT` and `MMT`, especially the advantage of sub-FL models in `MMT` for faster recovery after unlearning. Therefore, it is infeasible for large-scale experiments like CIFAR-100.

## 4.3 CONTINUAL LEARNING AND UNLEARNING

This experiment aims to simulate a continual setting in which new clients can join, and existing clients can leave the federation at any time during training. We will consider three settings corresponding to varying learning and unlearning order: 1) **2U1N**: Unlearn - New client - Unlearn; 2) **2U2N**: Unlearn - New client - Unlearn - New client; 3) **3U2N**: Unlearn - New client - Unlearn - New client - Unlearn. For a fixed number of communication rounds $k$, a new client will be introduced at round $k/2$, and an existing client will leave after every $k$ round. We use the same $k$ as the previous experiment. Fig. 5 shows performance in the continual learning and unlearning setting on the MNIST dataset. As can be seen, both `BMT` and `MMT` can seamlessly accommodate new clients and demonstrate general frameworks that can learn and unlearn rapidly.

---

[3] We did not include Exact-Fun baseline for CIFAR-100, as it has enormous GPU memory requirement to save all the intermediate model checkpoints, which is not feasible even with a H100 80GB GPU.

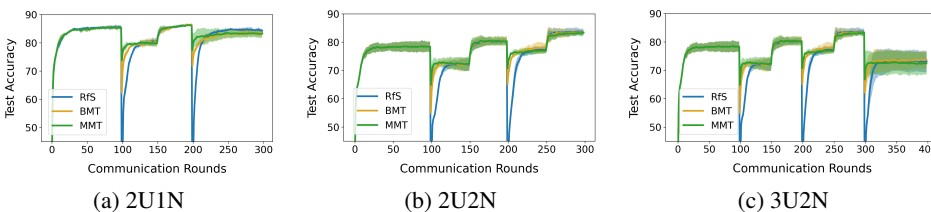

|  |  |  |
|:---:|:---:|:---:|
| (a) 2U1N | (b) 2U2N | (c) 3U2N |

Figure 5: Test accuracy in the continual learning and unlearning setting on MNIST.

## 4.4 ABLATION STUDIES

**Data heterogeneity.** As mentioned earlier, the data heterogeneity ratio $\rho$ defines the ratio between the number of samples in the majority and minority classes within a client's dataset. $\rho = 1$ indicates an IID dataset while $\rho \approx 0$ indicates an extremely non-IID dataset. As shown in Fig. 6, MMT consistently obtains the best performance across different heterogeneity ratios. The gap to other methods is more pronounced with lower $\rho$, suggesting MMT is more favorable when we have extremely non-IID data.

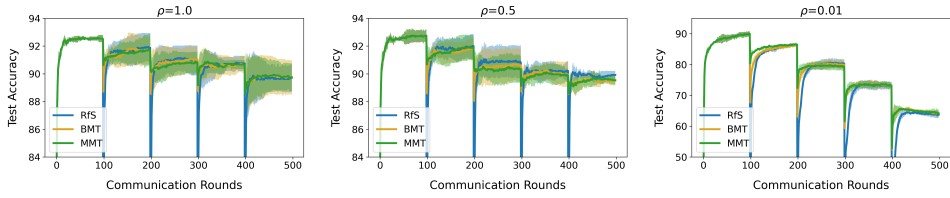

Figure 6: The effect of data heterogeneity on sequential unlearning performance on MNIST.

**Branching factor in MMT structure.** Recall that the default MMT uses a binary structure, i.e., a branching factor of $b = 2$ at each node in the tree of sub-FL modes. Therefore, we conduct experiments to analyze the impact of varying branching factors in MMT structure on the model performance. Particularly, for a federation consisting of $n$ clients, the branching factor can range from 1 to $n$ where $b = 1$ indicates a traditional setting with no sub-FL models while $b = n$ coincides with BMT, in which an auxiliary local model is maintained for each client. It is worth noting that we do not compare with $b = 1$ because it is equivalent to RfS.

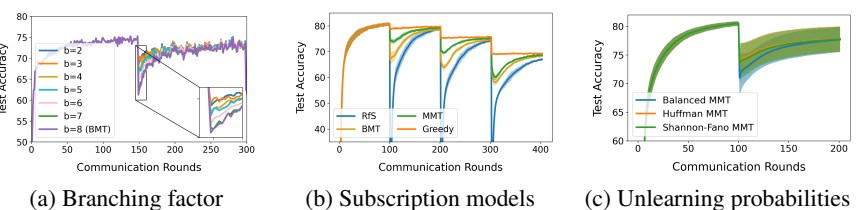

|  |  |  |
|:---:|:---:|:---:|
| (a) Branching factor | (b) Subscription models | (c) Unlearning probabilities |

Figure 7: **Left:** The effect of branching factor $b$ in MMT structure. **Middle:** Performance of greedily constructed MMT given fixed unlearning order. **Right:** Performance of various tree construction methods given non-uniform unlearning probabilities.

As can be seen in Fig. 7a, a smaller branching factor generally results in higher test accuracy. This improvement occurs because MMT with a smaller branching factor aggregates fewer sub-FL models during unlearning. Furthermore, each sub-FL model is more likely to converge to the global optimum due to training on more local datasets. Thus, MMT with the default binary structure is the most suitable configuration for unlearning.

**Subscription models.** The unlearning order can be fixed in certain circumstances, e.g. clients may subscribe to the service that allows them to participate in the federated process for a fixed duration and will leave once their subscription expires. In such scenarios, it is possible to construct an optimal MMT structure that maximizes learning performance when clients leave the federation in a fixed order. Particularly, the optimal structure is the one that arranges the clients by their expiration date, with the soon-to-expire client at the top and greedily building the tree until reaching those with the farthest expiration. As observed in Fig. 7b, the greedy implementation of MMT achieves the best unlearning

performance and outperforms the default `MMT` that assumes uniform probabilities of unlearning for all clients. Therefore, the greedy structure is preferable if the unlearning order of the clients is known in advance, e.g., in subscription models.

**Non-uniform unlearning probabilities.** The default `MMT` implementation assumes uniform unlearning probabilities for all clients. However, it is practical to consider the case where these probabilities are non-uniform. In fact, we will demonstrate that given the unlearning probabilities of each client, it is possible to construct improved `MMT` structures that achieve better unlearning performance through two strategies based on Shannon-Fano coding (Shannon, 1948) and Huffman coding (Huffman, 1952). Fig. 7c shows the performance for different tree construction methods. This result is obtained by sampling the client to be removed for 100 times according to pre-defined non-uniform unlearning probabilities of all clients. `MMT` structures that follow Shannon-Fano coding and Huffam coding obtain visibly improved results over the default `MMT`. Furthermore, Huffman-MMT obtains slightly better results than the Shannon-Fano counterpart, which aligns with the classical information theory results that Huffman coding is more optimal than Shannon-Fano coding for prefix-free code (Thomas & Joy, 2006).

## 4.5 Language Tasks

We also compare the performances of the proposed methods on two language tasks: 1) language identification, where the goal is to detect the language of the given text (Conneau, 2019), and 2) multilingual sentiment analysis, where the goal is to identify the sentiment of the given text using. We use the Huggingface papluca/language-identification dataset for the former task and Huggingface tyqiangz/multilingual-sentiments for the latter. We then randomly sample 200 and 500 data points separately for each class from top-8 classes with the most data. For both datasets, we finetune a pretrained GPT-2 Radford et al. (2019) model with the 200 data points set and Llama-3.2-3B model with the 500 data points set[4] to predict which language the input sequences belong to, with next-token prediction as the objective of getting the correct label. Fig 8 shows performance in unlearning settings for different NLP tasks. In all cases, `MMT` improves the fastest after unlearning, followed by `BMT`. In particular, for the larger model, both methods significantly outperformed other baselines, which corroborates our methods' scalability.[5] This experiment validates our method is effective across different modalities and model architectures.

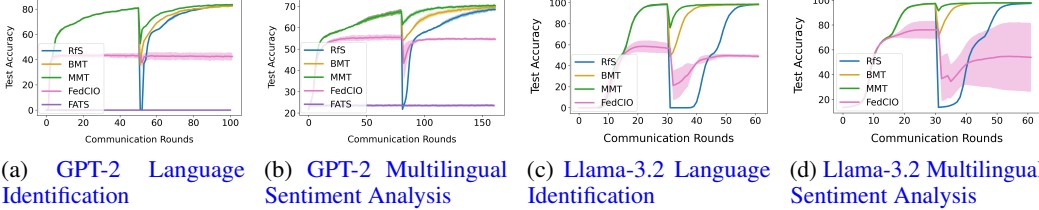

(a) GPT-2 Language Identification  (b) GPT-2 Multilingual Sentiment Analysis  (c) Llama-3.2 Language Identification  (d) Llama-3.2 Multilingual Sentiment Analysis

Figure 8: Sequential unlearning setting on two language tasks for GPT-2 and Llama-3.2-3B.

## 5 Conclusion

In this work, we propose two methods, `BMT` and `MMT`, for exact federated unlearning in the ongoing federated learning collaboration. Our methods ensure the complete removal of an unlearned client's data while having better performance post-unlearning with the remaining clients than retraining from scratch. Our methods are particularly useful in practical scenarios where model updation in collaborative environments cannot afford long delays, with minimal tolerance for interruptions. Our extensive experimental results demonstrate the effectiveness of the proposed methods. A few interesting future research directions include proposing a principal approach to design an influence tree under a resource constraint (i.e., the number of sub-FL models that can be trained is limited) and how to change the influence tree post-unlearning or after a client joins the collaboration while having the lowest IDS value and maximizing the use the existing trained sub-FL models.

---

[4]Due to the high computation and memory requirements to train multiple LLMs, we opted for GPT-2 and Llama-3.2-3B instead of the more popular and larger LLMs to show the performance of `BMT` and `MMT`.

[5]We did not include Exact-Fun for both LLM models and FATS for LLama-3.2-3B, as both methods have poor scalability w.r.t. model size due to their GPU memory requirement to save all intermediate model checkpoints.

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

## A    PROOF OF THEOREM 1

*Proof.* We first define the $k$-split influence node, which is a node in an influence tree with $k > 2$ leaf nodes. We first consider the influence tree $\mathcal{T}$, where $k$-split influence node only has leaf nodes as children. We denote this node as $d$, and the set of its leaf nodes is denoted by $\mathcal{C}$. We now follow the following procedure. First, we remove the edge between the node $d$ and any two of its leaf nodes (siblings), denoted by $i$ and $j$. We create a sub-FL model with these two removed nodes and then add the node for this sub-FL model as a child to the node $d$, as shown in Fig. 9. We denote the resulting tree as $\mathcal{T}'$. Let $f(\mathcal{T}, c)$ represent the number of sub-FL and local models that are aggregated to get the initial global modal after unlearning client $c$ in the given influence tree $\mathcal{T}$.

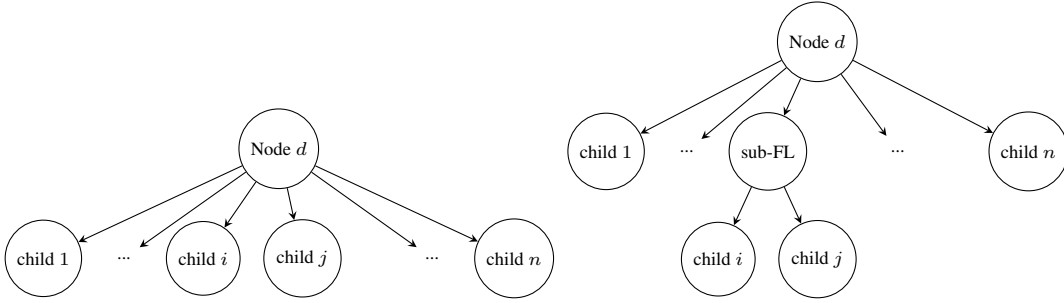

(a) Influence tree $\mathcal{T}$ before making any change.    (b) Influence tree $\mathcal{T}'$ after making the change.

Figure 9: Changes in influence tree structure.

Note that $f(\mathcal{T}', c) = f(\mathcal{T}, c) - 1$ for $c \in C \setminus \{i, j\}$ as one less leaf node to aggregate due to sub-FL model for $\{i, j\}$ leaf nodes, and $f(\mathcal{T}', c) = f(\mathcal{T}, c)$ for $c \in \{i, j\}$ as sub-FL model is no longer useful due to influence of leaf node $i$ or $j$. With this, we have following IDS due to the node $d$:

$$
\begin{aligned}
s_d(\mathcal{T}) = \sum_{c \in \mathcal{C}} p_c f(\mathcal{T}, c) &= \sum_{c \in \mathcal{C} \setminus \{i, j\}} p_c (f(\mathcal{T}', c) + 1) + \sum_{c \in \{i, j\}} p_c (f(\mathcal{T}', c) \\
&= k - 2 + \sum_{c \in \mathcal{C} \setminus \{i, j\}} p_c f(\mathcal{T}', c) + \sum_{c \in \{i, j\}} p_c (f(\mathcal{T}', c) \quad \text{(node $d$ had $k$ leaf nodes)} \\
&= k - 2 + \sum_{c \in \mathcal{C}} p_c f(\mathcal{T}', c) \\
&= s_d(\mathcal{T}') \\
\implies s_d(\mathcal{T}) &> s_d(\mathcal{T}'). \quad \text{(as $k > 2$)}
\end{aligned}
\tag{2}
$$

Iteratively apply the same procedure on the rest of the child nodes until every node only has two children. After this, we obtain a binary tree. Since each operation strictly reduces IDS, $s_d(\mathcal{T}_2) < s_d(\mathcal{T})$. When the original tree already has some child nodes $L$ that already belong to a binary subtree $\mathcal{T}_L$, we treat this subtree as a single child node $c'_k$ and apply the aforementioned operations on the child nodes that do not yet belong to a binary subtree. If all the child nodes belong to some binary subtree, we check from bottom-up to find the largest binary subtrees and treat them as a single child node to apply the aforementioned operations. Following this procedure, we can transform any arbitrary tree into a binary tree. In general,

$$
f(\mathcal{T}', l) = f(\mathcal{T}_L, l) + f(\mathcal{T}', c') = f(\mathcal{T}_L, l) + f(\mathcal{T}, c') - 1 = f(\mathcal{T}, l) - 1
$$

for $c' \in \mathcal{C} \setminus \{i, j\}, l \in L$ and

$$
f(\mathcal{T}', l) = f(\mathcal{T}_L, l) + f(\mathcal{T}', c') = f(\mathcal{T}_L, l) + f(\mathcal{T}, c') = f(\mathcal{T}, l)
$$

for $c' \in \{i, j\}, l \in L$. Notice that $f(\mathcal{T}', l) = f(\mathcal{T}_L, l) + f(\mathcal{T}', c')$ always holds. Therefore, the inequality in Eq. (2) generalizes for any tree structure with the generalized operation. After applying this procedure on any arbitrary tree $\mathcal{T}$ with at least one $k$-split influence node, the resulting binary tree $\mathcal{T}_2$ always has a strictly smaller value of IDS, i.e., $s(\mathcal{T}_2) < s(\mathcal{T})$.

Now we will proof the second part of theorem. Assume $N$ is the number of non-root nodes, $q_d$ is the probability of reaching a non-root node $d$ starting from the root node, and $N_d^s$ is the number of siblings of a non-root node $i$. Note that for a node $d$, $q_d = \sum_{c \in \mathcal{C}_d} p_c$ where $\mathcal{C}_d$ is the set of all the client nodes (i.e., leaf nodes) that are descendants of node $d$ and $p_c$ is the probability of unlearning of the $c$-th descendant. Given an influence binary tree $\mathcal{T}_2$ having $n$ clients with known unlearning probability of each client, the IDS is given as follows:

$$s(\mathcal{T}) = \sum_{c=1}^{n} p_c f(S, c) = \sum_{d=1}^{N} q_d * N_d^s. \tag{3}$$

For $\sum_{c=1}^{n} p_c f(S, c)$, we can group all leaf nodes that share some common ancestor node $d$ into a collection, with $\mathcal{C}_d$ denoting this set. Since the same node has the same $N_c^s$, we can sum $p_c$ of all such leaf nodes and rewrite $\sum_{c=1}^{n} p_c f(S, c)$ as $\sum_{d=1}^{N} \sum_{c \in \mathcal{C}_d} p_c * N_d^s = \sum_{d=1}^{N} q_d * N_d^s$. Since each node of the binary tree has only one sibling, we have

$$\sum_{d=1}^{N} q_d * N_d^s = \sum_{d=1}^{N} q_d * 1 = \sum_{d=1}^{N} \sum_{c \in \mathcal{C}_d} p_c = \sum_{c=1}^{n} p_c * l_c \tag{4}$$

where $n$ is the number of leaf nodes, $l_c$ is the depth of the $c$-th leaf node, and $p_c$ is the probability of reaching a non-root node $c$ starting from the root node. As splitting the $q_d$ of each non-leaf node into $\mathcal{C}_d = \{p_1, p_2, ..., p_\tau\}$, where $q_d = \sum_{c \in \mathcal{C}_d} p_c$ and each element in $\mathcal{C}_d$ corresponds to a $p_c$ of a leaf node, which is a descendant of that non-leaf node.

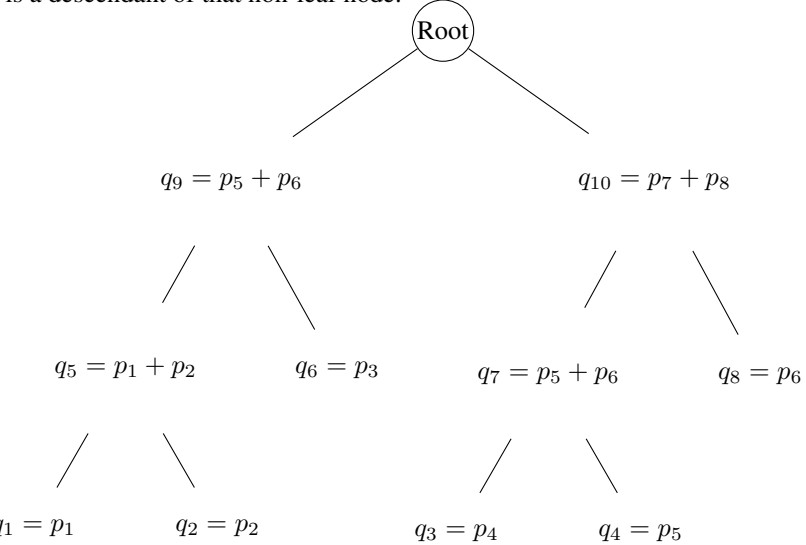

Figure 10: Visualization for $\sum_{d=1}^{N} q_d * 1 = \sum_{c=1}^{n} p_c * l_c$. One can easily see the equality holds.

Therefore, we can write $\sum_{d=1}^{N} q_d * 1$ as the sum of $p_c * l_c$ of all possible branches that reach each leaf node, as all the ancestor nodes of all leaf nodes have exactly one sibling (refer to Fig 10 for intuition). Finally, notice that $\sum_{c=1}^{n} p_c * l_c$ is the expected code word length, if the same binary tree is used to represent a binary prefix-free code encoding scheme. Since Huffman coding is optimal for minimizing $\sum_{c=1}^{n} p_c * l_c$, $s(\mathcal{T}_{\text{Huffman}}) \leq s(\mathcal{T}_2)$ where $\mathcal{T}_{\text{Huffman}}$ is an influence tree constructed following Huffman coding and $\mathcal{T}_2$ is any binary influence trees for the same set of $p_c$. $\qquad\square$

## B    ADDITIONAL EXPERIMENTAL RESULTS

**Benchmarking against SISA.**  SISA is a model-agnostic exact unlearning method proposed in Bourtoule et al. (2021). It involves partitioning the training dataset into disjoint subsets and training isolated models on each subset, whose predictions are aggregated during inference time. In our

context, we can train isolated models on each client's data and remove only the influenced model when a client leaves the collaboration to achieve exact unlearning. Fig. 11a shows SISA performance in the sequential unlearning setting on MNIST. Even though SISA obtains a better initialization than RfS, it incurs the worst post-unlearning performance compared to FL methods. This result suggests that the SISA training paradigm may not be well-suited for collaborative training among heterogeneous clients with limited data. Therefore, it serves as a less competitive exact FU benchmark.

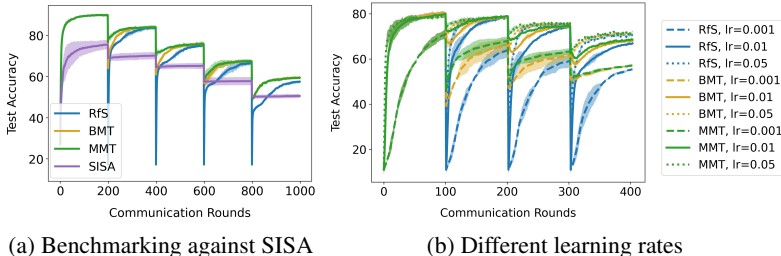

(a) Benchmarking against SISA      (b) Different learning rates

Figure 11: **Left:** Sequential unlearning benchmark against SISA. **Right:** Performance for different learning rates in the sequential unlearning setting.

**Varying learning rates.** Fig. 11b shows performance for three learning rates $\{0.001, 0.01, 0.05\}$ in the sequential unlearning setting on the MNIST dataset. In all cases, MMT converges the fastest, followed by BMT and RfS. This result validates the effectiveness of BMT and MMT compared to RfS across varying learning rates.

**Converged performance of CNN on CIFAR-100.** We increased the communication rounds from 500 to 1000 to analyze the converged performance of CNN trained on CIFAR-100. As seen in Fig. 12, there is a significant train-test accuracy gap for BMT and MMT starting from the 200th round after each unlearning request, indicating overfitting has occurred in both methods. This difference is due to using simple models like a 2-layer CNN when training for a long period. Therefore, we have included CIFAR-100 results on VGG-16 in Fig. 4(d).

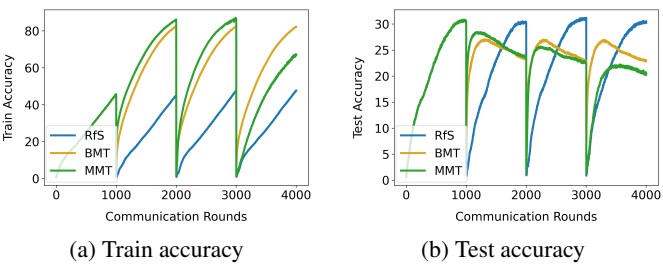

(a) Train accuracy      (b) Test accuracy

Figure 12: Performance at convergence on CIFAR-100 using a 2-layer CNN.

**Performance on unequal data.** In this experiment, we split the original dataset consisting of $n$ classes into $n$ clients with unequal data sizes. Specifically, each client transfers a portion $p \sim \mathcal{U}(0, 0.9)$ of their training data to another random client. We set the upper bound to 0.9 to prevent clients with empty data. As shown in Fig. 13, MMT and BMT consistently outperform RfS across all experiments, regardless of clients' data sizes.

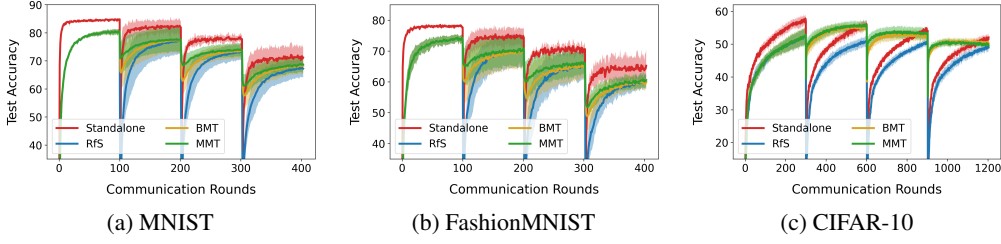

(a) MNIST      (b) FashionMNIST      (c) CIFAR-10

Figure 13: Performance on MNIST, FashionMNIST and CIFAR-10 with unequal clients' data.

