# OpenReview forum: "Achieving Exact Federated Unlearning with Improved Post-Unlearning Performance"
_ICLR.cc/2025/Conference — Submitted to ICLR 2025_

### Official Review · Reviewer_dvZG · 2024-11-03

**Soundness:** 2
**Presentation:** 3
**Contribution:** 2
**Rating:** 3
**Confidence:** 4

**Summary:**

This paper proposes an exact federated unlearning method which completely removes the influence of a particular client on the aggregated model. The traditional method for exact unlearning involves retraining the model from scratch. This is effective in removing influence, but also leads to degrading performance after unlearning. To address this issue, the authors propose to maintain a local model that is trained fully on local datasets. Instead of randomly reinitializing the model after unlearning, they use isolated local models for initialization. This method ensures exact unlearning while improving the post-unlearning performance.

**Strengths:**

1. The proposed algorithm effectively achieves exact unlearning by retraining from isolated local models instead of random reinitializations.
2. The method maintains model performance post-unlearning, making it highly practical for deployment in real-world federated systems.
3. The method is validated across multiple datasets, demonstrating its effectiveness in achieving better post-unlearning accuracy.

**Weaknesses:**

1. The core idea of this paper stems from using local models instead of random initialization. This trick is quite common in regular federated learning.
2. Definition 1 and theorem 1 seem redundant. Utilizing a disjoint binary tree structure to eliminate double influence is intuitive enough that it does not require additional justification or explanation.
3. The paper assumes that each local client has the same amount of data. However, varying data quantities can lead to significant differences in the performance of local models, which may impact post-unlearning aggregation. The authors should investigate this realistic scenario.
4. In figure 4(d), there seems to be a performance decline in BMT and MMT while the curve for retraining is still rising. Can the authors let the curve to converge and explain why this happens?

**Questions:**

Weaknesses.

---

> ### Author Response · Authors · 2024-11-24
> **Clarifications about local models and theoretical results,**
>
> Thank you for the detailed comments and suggestions. We also thank the reviewer for acknowledging that our proposed algorithms effectively achieve exact unlearning while maintaining model performance post-unlearning, which is also validated by our experiments on multiple datasets. We have responded to each of the concerns below.
>
>
> > **The core idea of this paper stems from using local models instead of random initialization. This trick is quite common in regular federated learning.**
>
> We partially agree with the reviewer that our core idea is to only use local models instead of random initialization. To achieve the performance gain in the initial rounds after unlearning, **MMT not only simultaneously trains local models (using only a single client's data) but also trains efficiently selected sub-FL models** (using the FL training process on a subset of clients' data) and then uses *these (local and sub-FL) models* to get a better initialized FL global model post-unlearning. Note that it is computationally infeasible to train sub-FL models for all subsets of clients as the number of clients increases; we introduced a notion of an **influence tree** to select the sub-FL models for training and then define a metric, **influence degradation score** that measures how good is the combination of selected sub-FLs models. We have proven that using Huffman coding with $n$ symbols representing clients and weights representing the clients' unlearning probability for building the influence tree gives the lowest influence degradation score. This result demonstrates how one can efficiently select these extra sub-FL models to get an FL initialization with a better performance in the initial rounds after unlearning. We have also corroborated our theoretical results using experiments on different vision and language datasets.
>
> Recent FL method, FedCIO *(Qiu et al., 2023)*, also achieves the exact federated unlearning using local model, however our experimental results (as shown in Fig. 4) show that FedCIO performs poorly as compared to BMT (that only uses local models) and MMT.
>
> 1. Qiu et al., 2023. FedCIO: Efficient Exact Federated Unlearning with Clustering, Isolation, and One-shot Aggregation. IEEE BigData.
>
>
> > **Definition 1 and theorem 1 seem redundant. Utilizing a disjoint binary tree structure to eliminate double influence is intuitive enough that it does not require additional justification or explanation.**
>
> We want to clarify that **Definition 1** (Definition 2 in the updated version) introduces a metric, **influence degradation score**, that measures how good the combination of selected sub-FLs models is in the  **influence tree**. We agree with the reviewer that a tree structure, by design, eliminates double influence (defined formally as Definition 1 in the updated version). However, we do not know the optimal tree structure given clients' different probabilities of requesting unlearning (as the chances of getting an unlearning request may vary across the clients). We answer this question in **Theorem 1** as we show that using Huffman coding with $n$ symbols representing clients and weights representing the clients' unlearning probability for building the influence tree gives the lowest *influence degradation score*. This result demonstrates how one can efficiently select these extra sub-FL models. Please refer to our updated Section 3.2 in the revised paper, highlighting the substantiated explanation.
>
>
> > **The paper assumes that each local client has the same amount of data. However, varying data quantities can lead to significant differences in the performance of local models, which may impact post-unlearning aggregation. The authors should investigate this realistic scenario.**
>
> Our proposed methods (BMT and MMT) and theoretical results do not need each local client to have the same amount of data. Since using the amount of data for each client is a common practice in FL experiments, we also use the same amount of data for each client in all our experiments. However, we also added the experimental results with different amounts of data for each client in the revised version of our paper (see Appendix B). As expected, BMT and MMT (also outperforming BMT) outperform the retraining from scratch (best baseline based on our experimental results in Fig. 4).

---

> ### Author Response · Authors · 2024-11-24
> **Clarifications about experiments**
>
> > **In figure 4(d), there seems to be a performance decline in BMT and MMT while the curve for retraining is still rising. Can the authors let the curve to converge and explain why this happens?**
>
> Thank you for your suggestion. We increased the communication rounds from 500 to 1000 to analyze the converged performance of CNN trained on CIFAR-100. Our new experimental results in Appendix B show a significant train-test accuracy gap for BMT and MMT starting from the 200th round after each unlearning request, indicating overfitting has occurred in both methods. This difference is due to using simple models like a 2-layer CNN when training for a long period. Therefore, we have included CIFAR-100 results on VGG-16 in Fig. 4(d) in the updated version of our paper. The new experimental results show that MMT and BMT outperform other baselines by a large margin.
>
> Thank you again for your time and your careful feedback. We hope our clarifications will alleviate your concerns and improve your opinion of our work.

---

> > ### Author Response · Authors · 2024-11-30
> >
> > Dear Reviewer dvZG,
> >
> >    \
> > Thank you once again for reviewing our paper and providing thoughtful and valuable feedback. We hope our rebuttal has addressed your concerns. With the discussion period approaching its conclusion, please let us know if there are any additional concerns and questions that we can address. If you feel our responses have sufficiently resolved your concerns, we would greatly appreciate it if you could consider re-evaluating your score.
> >
> >    \
> > Best,\
> > Authors

---

### Official Review · Reviewer_XuT7 · 2024-11-04

**Soundness:** 2
**Presentation:** 2
**Contribution:** 2
**Rating:** 5
**Confidence:** 5

**Summary:**

In this paper, the authors propose an exact unlearning method for federated model training. The main idea behind this paper is to better initialize the global model and re-train the global model on the remaining clients in a federated way. The authors further propose two strategies for the global model initialization in federated unlearning, including using the first-round local models of remaining clients and using the first-round local models and the corresponding sub-FL models of the remaining clients. Experiments on several datasets show the proposed method can outperform a naive baseline.

**Strengths:**

1. The federated unlearning studied in this paper is an important research problem.
2. This paper is well-written and easy to follow.

**Weaknesses:**

1. The academic findings found in this paper are not novel.
The main academic findings brought by this paper are that using the proposed initialization strategies (using some locally trained models) rather than random initialization can improve the efficiency of federated exact unlearning. However, it is straightforward that using some locally trained models can speed up the convergence of the federated exact unlearning.

2. The proposed initialization algorithm, i.e., MMT, is not well-motivated.
(1) In lines 241-253, the authors claim that we should capture the joint influence of multiple clients for better global model initialization. However, the authors do not give any clear definition of the client's joint influence. To the best of my knowledge, "client joint influence" is not common knowledge in the area of federated learning.
(2) In lines 263-269, the authors claim that "If the number of models to aggregate is less, it implies that the initialization of the global model contains the most joint influence of clients". It is also difficult to understand this statement. If we only use a small part of local models for global model initialization, why can we better capture the client joint influence?

3. The MMT method is inefficient in both computation and storage.
In the proposed MMT method, the authors introduce a lot of sub-models to memorize the historical training state. However, this will introduce many extra computation costs. For example, if there exists $n = 2^m$ clients, and we maintain an influence tree with $h$ levels (including the top level and the bottom level), the computation costs of the FL systems will increase $h$ times, which may be impractical in real-world applications.

4. The experiments in this paper can be improved.
(1) Only small-scale datasets are used for experiments, i.e., MNIST, FMNIST, and CIFAR. The authors should compare different methods on larger datasets to demonstrate the superiority of the proposed method.
(2) Only small-scale models, i.e., two-layer MLPs and two-layer CNNs, are used for experiments. Can the proposed method be applied to large-scale models, especially for the LLMs?
(3) The authors only compare a trivial baseline method, i.e. training from scratch. However, there are many SOTA federated unlearning methods, the authors should compare them in this paper.

**Questions:**

Refer to weakness.

---

> ### Author Response · Authors · 2024-11-24
> **Clarifications about contributions and computation costs**
>
> We would like to thank you for the detailed comments and suggestions. We also thank the reviewer for acknowledging that the problem we are studying is an important research problem and that our paper is well-written and easy to follow. We have responded to each of the concerns below.
>
>
> > **The academic findings found in this paper are not novel. The main academic findings brought by this paper are that using the proposed initialization strategies (using some locally trained models) rather than random initialization can improve the efficiency of federated exact unlearning. However, it is straightforward that using some locally trained models can speed up the convergence of the federated exact unlearning.**
>
> We agree with the reviewer that our methods are straightforward, and hence, they can be easily incorporated with any existing FL algorithm. We have proposed two methods (BMT and MMT) for federated learning that allow exact unlearning while having better post-unlearning performance. To achieve this performance gain, we simultaneously train multiple models using only a single client's data (local model) or a subset of clients' data (**sub-FL model, which is also trained using FL training process**) and then use them to get a better FL global model post-unlearning. Training these additional models in parallel to the global FL model adds extra computational overhead but helps to achieve good post-unlearning performance, which may be more desirable in many real-life applications. Note that it is computationally infeasible to train sub-FL models for all subsets of clients as the number of clients increases; we introduced a notion of an **influence tree** to select the sub-FL models for training and then define a metric, **influence degradation score** that measures how good is the combination of selected sub-FLs models. We have proven that using Huffman coding with $n$ symbols representing clients and weights representing the clients' unlearning probability for building the influence tree gives the lowest influence degradation score. This result demonstrates how one can efficiently select these extra sub-FL models to get an FL initialization with a better performance in the initial rounds after unlearning. We have also corroborated our theoretical results using experiments on different vision and language datasets.
>
>
> > **In lines 241-253, the authors claim that we should capture the joint influence of multiple clients for better global model initialization. However, the authors do not give any clear definition of the client's joint influence. To the best of my knowledge, "client joint influence" is not common knowledge in the area of federated learning.**
>
> **Client Joint influence** refers to the overall influence of the clients on the model that uses these clients' data. For example, the *local models* are only trained on the individual client's data; hence, these models are not influenced by other clients' data. Whereas *sub-FL models* are trained in a subset of clients' data; therefore, these clients will jointly influence the model via multiple FL rounds by these clients' data. Due to the joint influence, the sub-FL model is always better than a model that is a weightage average of local models. Therefore, using sub-FL models leads to a better initialization of the global FL model after unlearning.
>
> > **In lines 263-269, the authors claim that "If the number of models to aggregate is less, it implies that the initialization of the global model contains the most joint influence of clients". It is also difficult to understand this statement. If we only use a small part of local models for global model initialization, why can we better capture the client joint influence?**
>
> After receiving an unlearning request, we used the remaining *uninfluenced local and sub-FL models* to get the re-initialization for the global FL model in MMT. Since we only use one model (sub-FL or local) for each client, we first want to use a sub-FL model that is influenced by a maximum number of clients' data that captured the influence of interactions of clients' heterogeneous data via FL updates on the model and then consider the next biggest sub-FL model for the remaining clients, and so on. For example, if there is a sub-FL model for clients {5,6,7,8}, then we use this sub-FL model over the sub-models trained on the data of client {5,6} and {7,8}, as shown in Fig. 3(c). This process allows us to use the maximal possible joint influence while considering each client once. Therefore, if the number of models (local and sub-FL) used to get the initialization of the global FL model is less, then we are using sub-FL models having higher joint influence (trained on more clients' data), leading to better initialization. We have corroborated this with our experimental results, showing MMT works better than BMT.

---

> ### Author Response · Authors · 2024-11-24
> **More clarifications about computation costs**
>
> > **The MMT method is inefficient in both computation and storage. In the proposed MMT method, the authors introduce a lot of sub-models to memorize the historical training state. However, this will introduce many extra computation costs.**
>
> We agree with the reviewer that the local computing and communication costs can be higher for MMT than for standard FL learning due to training the multiple sub-FL and local models. However, MMT re-initializing the global model using disjoint sub-FL models is advantageous, as it retains and leverages more useful information from the remaining clients, leading to better performance, as shown in our empirical results. Modern infrastructure can be optimized for these aspects; specifically, the $n-1$ sub-FL models needed can be trained in parallel to mitigate computation delay. The storage of sub-FL models can be distributed amongst clients, resulting in only a marginal increase in memory usage at the client level. Furthermore, note that existing exact FL unlearning methods like Exact-Fun *(Xiong et al., 2023)* and FATS *(Tao et al., 2024)* require storing all intermediate checkpoints to perform unlearning, which scales linearly w.r.t. the training time and this scales poorly w.r.t. model and training data size, whereas our method does not incur more training and storage cost w.r.t. these factors, and we show our methods even perform better on larger models in the updated NLP tasks (see Fig. 8(c) and 8(d) in the updated version of our paper).
>
> 1. Xiong et al., 2023. Exact-Fun: An Exact and Efficient Federated Unlearning Approach. ICDM.
> 2. Tao et al., 2024. Communication efficient and provable federated unlearning. arXiv:2401.11018.
>
> The additional computing and communication costs are the price of having an exact unlearning guarantee and a model with a better performance in the initial rounds after unlearning. In many real-world applications where the unlearning efficiency (i.e., how fast the unlearned model can be used again in deployment) is critical, it is worth frontloading extra computational and storage costs. Therefore, we can adopt methods like MMT in practical settings to have the exact unlearning guarantee with improved post-unlearning performance compared to retraining from scratch, which uses a randomly initialized model after each unlearning request.
>
>
> > **if there exists $n=2^m$ clients, and we maintain an influence tree with $h$ levels (including the top level and the bottom level), the computation costs of the FL systems will increase $h$ times, which may be impractical in real-world applications.**
>
> We want to clarify that the overall computation costs of the FL systems will increase linearly with the number of clients ($n$). Instead, the additional models MMT needs is $2n-2$ as there will be $n$ local models (one for each client) and $n-2$ sub-FL models as the binary tree is the optimal influence tree. For a balanced binary influence tree, the computational cost scales logarithmically with respect to the number of clients (depth of the influence tree). Since we only focus on cross-silo FL settings where the number of clients typically does not exceed a few hundred, this computational cost will only increase a few times (i.e., $\log(n)$) than standard FL systems.

---

> ### Author Response · Authors · 2024-11-24
> **Clarifications about experiments**
>
> > **The experiments in this paper can be improved. (1) Only small-scale datasets are used for experiments, i.e., MNIST, FMNIST, and CIFAR. The authors should compare different methods on larger datasets to demonstrate the superiority of the proposed method. (2) Only small-scale models, i.e., two-layer MLPs and two-layer CNNs, are used for experiments. Can the proposed method be applied to large-scale models, especially for the LLMs?**
>
> Our methods are evaluated not only in small-scale settings but also in large-scale ones with LLMs (**GPT-2**). Specifically, Fig. 8(a) and 8(b) in our paper show that MMT and BMT outperform RfS (retraining from scratch) by a large margin for GPT-2 models on two language tasks. Additionally, we updated new results for large language models in Fig. 4(d) (**VGG-16 on CIFAR-100**) and Fig. 8(c) and 8(d) (**Llama-3.2-3B on two language tasks**). In all cases, MMT has the best performance (i.e., high accuracy) in the initial rounds after unlearning, followed by BMT. Specifically, both of our proposed methods significantly outperformed the retraining from scratch baseline for the larger models, which corroborates our methods' scalability.
>
>
> > **The authors only compare a trivial baseline method, i.e. training from scratch. However, there are many SOTA federated unlearning methods, the authors should compare them in this paper.**
>
> In the revised version of our paper, Fig. 4 has included the comparisons with the SOTA federated unlearning method that guarantees some form of exact unlearning: Exact-Fun *(Xiong et al., 2023)*, FedCIO *(Qiu et al., 2023)*, and FATS *(Tao et al., 2024)*.
>
> 1. Xiong et al., 2023. Exact-Fun: An Exact and Efficient Federated Unlearning Approach. ICDM.
> 2. Qiu et al., 2023. FedCIO: Efficient Exact Federated Unlearning with Clustering, Isolation, and One-shot Aggregation. IEEE BigData.
> 3. Tao et al., 2024. Communication efficient and provable federated unlearning. arXiv:2401.11018.
>
> In all our experiments, BMT and MMT consistently outperform existing FL methods by a large margin with better initialization and performance (i.e., high accuracy) in the initial rounds after unlearning.
>
>
>
> Thank you again for your time and your careful feedback. We hope our clarifications will alleviate your concerns and improve your opinion of our work.

---

> > ### Comment · Reviewer_XuT7 · 2024-11-28
> >
> > Thank you for your detailed response. I have revised my score.

---

> > > ### Author Response · Authors · 2024-11-28
> > >
> > > Thank you for increasing your score.

---

### Official Review · Reviewer_4u7k · 2024-11-04

**Soundness:** 2
**Presentation:** 2
**Contribution:** 1
**Rating:** 3
**Confidence:** 4

**Summary:**

This paper tackles the challenge of achieving exact federated unlearning while maintaining good post-unlearning performance. Existing methods, such as retraining the federated model from scratch, fail to provide satisfactory initial performance after unlearning. Therefore, the authors propose  Bi-Models Training (BMT) and Multi-Models Training (MMT) to address this issue. BMT preserves isolated copies of local models and reuses clients' existing knowledge during unlearning. MMT trains multiple sub-federated learning models on disjoint subsets of clients and aggregates the best sub-models upon unlearning. Both methods ensure exact federated unlearning while achieving improved performance compared to retraining from scratch.

**Strengths:**

**S1**. The topic of how to simultaneously preserve the model performance while exactly removing one client's influence is an important problem in practice.

**S2**. Both theoretical and empirical results were provided to verify the effectiveness of the proposed method.

**Weaknesses:**

**W1**. The technical contributions of the proposed methods are limited. For BMT, it only re-initializes the global model with local models that are only trained once by remaining clients, which is only equal to saving one round's communication cost compared to the random initialization case. For MMT, it maintains multiple series of model training (e.g., global model training, sub-FL training, and local training) to increase the training process's robustness to clients' exclusion. The local computation and communication cost can be larger than restart training the model from the randomly initialized one.

**W2**. Some claims are confusing. For example, the reasons for the new model's low accuracy need further clarification in line 161 "the new model may have very low accuracy compared to the aggregated model before unlearning due to restarting the FL process with random initialization ", since restarting the FL process with remaining clients should not cause performance drop given unlimited the communication and computation resources.

**W3**. The definition of *double influence* and its impact are also unclear. Please clarify this term in the response.

**Questions:**

See the weakness.

---

> ### Author Response · Authors · 2024-11-24
> **Clarifications about technical contributions and computation costs**
>
> We would like to thank you for the detailed comments and suggestions. We also thank the reviewer for acknowledging that the problem we are studying is important. We have responded to each of the concerns below.
>
> > **The technical contributions of the proposed methods are limited.**
>
> Our technical contributions are summarized as follows:
> 1. We have proposed new methods (BMT and MMT)  for federated learning that allow exact unlearning while having better post-unlearning performance.
>
> 2. To achieve this performance gain, we simultaneously train multiple models using only a single client's data (local model) or a subset of clients' data (sub-FL model) and then use them to get a better FL global model post-unlearning. Training these additional models in parallel to the global FL model adds extra computational overhead but helps to achieve good post-unlearning performance with very short unlearning and recovery time, which is more desirable in many real-life applications like machine learning models used to assist disease diagnosis, credit score prediction models, and fine-tuned LLMs in domains like healthcare and banking.
>
> 3. Since it is computationally infeasible to train sub-FL models for all possible subsets (i.e., all elements of the clients' power set) of clients as the number of clients increases, we introduced a notion of an **influence tree** to select the sub-FL models for training and then define a metric, **influence degradation score** that measures how good is the combination of selected sub-FLs models.
>
> 4. We have proven that using Huffman coding with $n$ symbols representing clients and weights representing the clients' unlearning probability for building the influence tree gives the lowest influence degradation score. This result demonstrates how one can efficiently select these extra sub-FL models to get an FL initialization with a better performance in the initial rounds after unlearning. We have also corroborated our theoretical results using experiments on different vision and language datasets.
>
>
>
> > **For BMT, it only re-initializes the global model with local models that are only trained once by remaining clients, which is only equal to saving one round's communication cost compared to the random initialization case.**
>
> BMT re-initializes the global model after receiving the unlearning request using only the remaining clients' local models that, by design, remove the influence of the clients to be unlearned. The re-initialized FL model using BMT is better than a randomly initialized one, as the local models are already trained on the clients' local datasets. The advantage of BMT becomes more evident when there are multiple unlearning requests from different clients, and the local models are well-trained (having high accuracy) on the clients' local datasets.
>
> > **For MMT, it maintains multiple series of model training (e.g., global model training, sub-FL training, and local training) to increase the training process's robustness to clients' exclusion. The local computation and communication cost can be larger than restart training the model from the randomly initialized one.**
>
> We agree with the reviewer that the local computing and communication costs can be higher for MMT than for standard FL learning. This extra cost is due to training the multiple sub-FL and local models in MMT. However, MMT re-initializing the global model using disjoint sub-FL models is advantageous, as it retains and leverages more useful information from the remaining clients, leading to better performance, as shown in our empirical results. Modern infrastructure can be optimized for these aspects; specifically, $n-1$ sub-FL models can be trained in parallel to mitigate computation delay. The storage of sub-FL models can be distributed amongst clients, resulting in only a marginal increase in memory usage at the client level. Furthermore, note that existing exact FL unlearning methods like Exact-Fun *(Xiong et al., 2023)* and FATS *(Tao et al., 2024)* require storing all intermediate checkpoints to perform unlearning, which scales linearly with training time and has poor scalability for larger models and datasets. These additional computing and communication costs are the price of having an exact unlearning guarantee and a model with a better performance in the initial rounds after unlearning. Therefore, we can adopt methods like MMT in practical settings to have the exact unlearning guarantee with improved post-unlearning performance compared to retraining from scratch, which uses a randomly initialized model after each unlearning request.
>
> 1. Xiong et al., 2023. Exact-Fun: An Exact and Efficient Federated Unlearning Approach. ICDM.
> 2. Tao et al., 2024. Communication efficient and provable federated unlearning. arXiv:2401.11018.

---

> ### Author Response · Authors · 2024-11-24
> **Clarifications about computation costs and double influence**
>
> > **the reasons for the new model's low accuracy need further clarification in line 161 "the new model may have very low accuracy compared to the aggregated model before unlearning due to restarting the FL process with random initialization ", since restarting the FL process with remaining clients should not cause performance drop given unlimited the communication and computation resources.**
>
> To clarify, the **new model** in line 161 refers to the FL model just after the unlearning request, e.g., the randomly initiated FL model is retraining from scratch. This new FL model may have very low accuracy **in the initial FL rounds after unlearning** compared to the aggregated model before unlearning due to restarting the FL process with random initialization. The service providers using the FL models in many real-life applications like healthcare or finance must provide their services around the clock. Therefore, after each unlearning request, they can not wait for the FL model to converge before restarting their services, as it may take time for FL re-training (e.g., fine-tuning LLMs ). Therefore, we need methods to get better FL models in the initial rounds post-unlearning. Our proposed methods, BMT and MMT, fill this gap and allow us to get better FL models post-unlearning than other existing FL methods like retraining from scratch, as shown in our updated experiment results.
>
>
> > **The definition of double influence and its impact are also unclear. Please clarify this term in the response.**
>
> The following is the formal definition of the **double influence**, which we have also added in the revised version of the paper:
> > **Double Influence:** Let $S_i$ be the set of clients whose data are used in training the $i$-th sub-FL model. Then, sub-models $i$ and $j$ leads to double influence iff $S_i \cap S_j  \ne \emptyset$, $S_i \setminus S_j \ne \emptyset$, and $S_j \setminus S_i \ne \emptyset$.
>
> In simple terms, when one client data is used to train two sub-FL models, it can lead to double influence if both are also trained using data from different clients, e.g., one is trained on clients {$1,2$}, while another on clients {$1,3$}; the data of client $1$ is used in both sub-FL models and hence having the double influence. We can not aggregate the sub-FL models with double influence via any standard FL protocol (e.g., the weighted averaging).
>
>
> Thank you again for your time and your careful feedback. We hope our clarifications will alleviate your concerns and improve your opinion of our work.

---

> > ### Author Response · Authors · 2024-11-30
> >
> > Dear Reviewer 4u7k,
> >
> >    \
> > Thank you once again for taking the time to review our paper and provide thoughtful and valuable feedback. We hope that our rebuttal has addressed your concerns. As the discussion period approaches its conclusion, we kindly ask if we can address any additional concerns or questions. If you think that our responses have sufficiently addressed your concerns, we would be truly grateful if you could consider re-evaluating your score.
> >
> >    \
> > Best,\
> > Authors

---

### Meta-Review · Area_Chair_YJcL · 2024-12-19

**Metareview:**

The paper studies unlearning from an FL standpoint. The authors propose two new methods for federated unlearning, that require training multiple models per subsets of clients, which is made efficient though the construction of a hierarchy of influence across client subsets, identifying how to group them together. This is shown to be performant w.r.t. to unlearning but also w.r.t. post unlearning predictions via experiments.

Some reviewers recognized strengths in the importance of the problem and in the experiments. Including larger datasets both for vision and text during the rebuttal stage improved the paper, and the diversity of datasets was considered a strength.

Issues were raised regarding the technical novelty of the paper: techniques were considered straightforward heuristics. Both the motivation of the proposed method and its overall efficiency were brought into question.

**Additional Comments On Reviewer Discussion:**

Reviewers remained concerned regarding the technical novelty and the efficiency of the proposed method.

---

### Decision · Program_Chairs · 2025-01-22

Reject